# Nutritional Evaluation of Sea Buckthorn "*Hippophae rhamnoides*" Berries and the Pharmaceutical Potential of the Fermented Juice

**Sobhy A. El-Sohaimy** [1,2,*] **, Mohamed G. Shehata** [1,3] **, Ashwani Mathur** [4] **, Amira G. Darwish** [1] **, Nourhan M. Abd El-Aziz** [1] **, Pammi Gauba** [4] **and Pooja Upadhyay** [4]

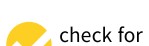



1 Department of Food Technology, Arid Lands Cultivation Research Institute, City of Scientific Research and Technological Applications, New Borg El Arab 21934, Alexandria, Egypt
2 Department of Technology and Organization of Public Catering, Institute of Sport Tourism and Services, South Ural State University, 454080 Chelyabinsk, Russia
3 Food Research Section, R&D Division, Abu Dhabi Agriculture and Food Safety Authority (ADAFSA), Abu Dhabi P.O. Box 52150, United Arab Emirates
4 Department of Biotechnology, Jaypee Institute of Information Technology Noida, A-10, Sector-62, Noida 201309, Uttar Pradesh, India
* Correspondence: alsukhaimisa@susu.ru or selsohaimy@srtacity.sci.eg

**Abstract:** Sea buckthorn is a temperate bush plant native to Asian and European countries, explored across the world in traditional medicine to treat various diseases due to the presence of an exceptionally high content of phenolics, flavonoids and antioxidants. In addition to the evaluation of nutrients and active compounds, the focus of the present work was to assess the optimal levels for *L. plantarum* RM1 growth by applying response surface methodology (RSM), and to determine the impact of juice fermentation on antioxidant, anti-hypertension and anticancer activity, as well as on organoleptic properties. Sea buckthorn berries were shown to contain good fiber content (6.55%, 25 DV%), high quality of protein (3.12%, 6.24 DV%) containing: histidine, valine, threonine, leucine and lysine (with AAS 24.32, 23.66, 23.09, 23.05 and 21.71%, respectively), and 4.45% sugar that provides only 79 calories. Potassium was shown to be the abundant mineral content (793.43%, 22.66 DV), followed by copper and phosphorus (21.81 and 11.07 DV%, respectively). Sea buckthorn juice exhibited a rich phenolic, flavonoid and carotenoid content (283.58, 118.42 and 6.5 mg/g, respectively), in addition to a high content of vitamin C (322.33 mg/g). The HPLC profile indicated that benzoic acid is the dominant phenolic compound in sea buckthorn berries (3825.90 mg/kg). Antioxidant potentials (DPPH and ABTS) of sea buckthorn showed higher inhibition than ascorbic acid. Antimicrobial potentials were most pronounced against *Escherichia coli* BA12296 (17.46 mm). The probiotic growth was 8.5 log cfu/mL, with juice concentration, inoculum size and temperature as the main contributors to probiotic growth with a 95% confidence level. Fermentation of sea buckthorn juice with *L. plantarum* RM1 enhanced the functional phenolic and flavonoid content, as well as antioxidant and antimicrobial activities. The fermentation with *L. plantarum* RM1 enhanced the anti-hypertension and anticancer properties of the sea buckthorn juice and gained consumers' sensorial overall acceptance.

**Keywords:** sea buckthorn juice; *L. plantarum* RM1; fermentation; response surface methodology (RSM); antioxidant; antimicrobial; Angiotensin Converting Enzyme (ACE) inhibition; Caco-2; HT-22 cell line

## 1. Introduction

Sea buckthorn (*Hippophae rhamnoides* L., subsp. *carpatica*) is an ancient crop that was traditionally used as herbal medicine, a health promotive and a food additive. Recently, the plant and its derivatives have gained worldwide attention due to presence of different kinds of nutrients and bioactive compounds such as vitamins, amino acids, carotenoids

and phenolic compounds present in parts of the plant, including the berries, with nutritional and medicinal values [1–4]. Sea buckthorn has been reported to have a wide range of activities from lowing cholesterol, platelet aggregation, blood pressure and blood sugar to anticancer, antifungal, antibacterial, antihistaminic, antiviral, spasmolytic and radioprotective potential [5]. Fermentation of sea buckthorn juice with lactic acid bacteria (LAB) presents a potential approach to modify the contents of key compounds of berry composition, while also contributing to the unwanted sensory properties through malolactic fermentation (MLF). This process involves decarboxylation of malic acid to lactic acid that reduces acidity, increasing microbiological stability, and modifying the aroma, flavor and texture of the juice. Among different lactic acid bacteria, *Lactobacillus* sp. showed high efficiency and adaptation according to the available literature [6]. Fermentation of the juice have been documented to increase antioxidant and antimicrobial activities, and consequently result in increased functional benefits [7].

According to world statistics, twenty percent of the adult population is suffering from hypertension and related diseases such as coronary heart disease (CHD), kidney dysfunction and myocardial infarction [8]. Hypertension is known to be regulated by the activity of a zinc protease enzyme, called angiotensin-converting enzyme (ACE) [9]. ACE catalyzes the production of potent vasoconstrictor angiotensin-II from inactive angiotensin-I and inactivates the vasodilator peptide bradykinin, thus playing a crucial role in blood pressure regulation [10,11]. ACE inhibitors, commonly available in markets and which include captopril, enalapril, alacepril or lisinopril, are often used to treat various cardio-related diseases [12]. Due to certain side-effects (inflammatory response, taste disturbance, dry cough or angioneurotic edema), ACE inhibitors from protein hydrolysates are gaining attention as potential alternatives to antihypertensive drugs [13]. In various studies, compounds present in sea buckthorn plays a role in inhibiting the ACE activity. The objective of the current work was the evaluation of the nutritional, functional and therapeutic properties (anti-hypertension and anticancer) of sea buckthorn juice fermented using *L. plantarum* RM1. The growth of the probiotic organism was optimized using response surface methodology (RSM). Furthermore, the organoleptic properties were evaluated pre- and post-fermentation with the probiotic bacteria.

## 2. Materials and Methods

### 2.1. Materials and Culture Preparation

Lyophilized Sea buckthorn berries (*Hippophae rhamnoides* L., ssp. *Carpatica*) were purchased from a local Indian market via the Amazon website (https://www.amazon.com, accessed on: 17 October 2022). According to the trader statement, the berries originated from India. The berries were kept in plastic bags, directly refrigerated at 4 °C/8 h, and subsequently frozen and stored at −20 °C. The compounds 2,2-diphenyl-1-picrylhydrazyl (DPPH), 2,2′-azino-bis (3-ethylbenzothiazoline-6-sulfonic acid) (ABTS), 3,4,5-trihydroxybenzoic acid (gallic acid) and β-carotene were obtained from Sigma Chemical Co. (St. Louis, MO, USA). Folin–Ciocalteu reagent, sodium carbonate, sodium hydroxide and ethanol (HPLC grade) were obtained from Loba Chemie (Mumbai, India). All solvents/reagents used were of analytical grade. *Lactobacillus plantarum* RM1 (MF817708) was isolated by us in the Department of Food Technology, City of Scientific Research, from fermented milk (Rayeb milk) that was purchased from a local producer in Alexandria, Egypt. Strain RM1 was cultivated in De Man Rogosa and Sharpe broth (MRS) (Conda, Spain) at 37 °C/24 h. For a long storage, strain RM1 was stored in 30% glycerol at −20 °C.

### 2.2. Chemical Composition

The chemical composition: pH, Brix and titratable acidity of the sea buckthorn berries were measured as described by Araya-Farias et al. [14] and Shehata et al. [15]. The sugar content was determined as glucose by the phenol-sulfuric method [16] using the standard curve of gradient concentrations of glucose.

### 2.3. Amino Acids Composition of Sea Buckthorn

Amino acid analysis was carried out by Automatic Amino Acid Analyzer (AAA 400 INGOS Ltd., Prague, Czech Republic) using the performic acid oxidation method according to INGOS [17] and Smith [18]. The enquired pattern values were according to FAO/WHO/UNU [19], FAO/WHO [20] and Recommended Dietary Allowances, Food and Nutrition Board, Commission on Life Sciences, National Research Council [21], and amino acid score (AAS%) was calculated as "percentage of adequacy" as follows:

$$\text{Amino Acid Score } (\%) = \frac{\text{mg of amino acid in 1 g test protein}}{\text{mg of amino acid in requirement pattern}} \times 100 \quad (1)$$

### 2.4. Extraction and Sample Preparation

Lyophilized samples of sea buckthorn berries (1.0 g) were extracted and dissolved in 10 mL of water in a shaking water bath (Daihan Scientific Co., Ltd., Bangkok, Thailand) at 50 °C/60 min. The mixture was centrifuged at $2430 \times g$/15 min (Pro-Research, Centurion Scientific Ltd., Chichester, UK). The extract was lyophilized for 48 h/$-56$ °C in a Dura-Dry MP freeze dryer (Model FDF 0350, Gyeonggi-do, Korea) at 0.04 Mbar and stored at $-20$ °C until used.

Vitamin C in lyophilized samples of berries (0.5 g) was extracted according to Abdel-Razek et al. [22] in 25 mL extraction mixture of methanol/$H_3PO_4$/redistilled water (99/0.5/0.5, *v/v/v*) in shaker (Daihan Scientific Co., Ltd., Bangkok, Thailand)) for 10 min in the dark. Individual carotenoids were isolated from lyophilized samples of berries (1.0 g) using 10 mL extraction mixture (hexane/acetone/ethanol in the ratio of 99/0.5/0.5, *v/v/v*) in a shaker for 30 min in the dark. The upper layer of hexane (1 mL) was dried and dissolved in 1 mL mixture of acetonitrile/methanol in the ratio of 30:55 (*v/v*).

### 2.5. Optimization of Variables of Probiotic Growth Using Response Surface Methodology

The response surface methodology (RSM) was employed for the identification of significant variables for enhanced growth of probiotics in sea buckthorn juice [23]. In the current study, RSM was carried out using central composite design (CCD). CCD contains 3 different levels of each factor. In this experiment, 20 experimental trials were carried out with 3 independent variables, which are represented as A (juice concentration), B (inoculum size) and C (temperature) at 3 coded levels (+1, 0, −1). To interpret the results, a second order polynomial equation was employed, interpreting the correlation between the variables and the response [24,25]. The second-degree polynomial equation is as follows:

$$\left( Y = \beta_0 + \beta_1 X_1 + \beta_2 X_2 + \beta_3 X_3 + \beta_{11} X_{12} + \beta_{22} X_2^2 + \beta_{33} X_3^2 + \beta_{12} X_1 X_2 + \beta_{13} X_1 X_3 + \beta_{23} X_2 X_3 \right) \quad (2)$$

where $Y$ is the predicted response, $\beta_0$ is the model constant, $\beta_1$, $\beta_2$ and $\beta_3$ are the linear co-efficients, $\beta_{11}$, $\beta_{22}$ and $\beta_{33}$ are the squared co-efficients, and $\beta_{12}$, $\beta_{13}$ and $\beta_{23}$ the interaction co-efficients.

All designs and calculations were carried out by Design Expert Software (Version 12.0.1.0, State-Ease, Minneapolis, MN, USA). The results were analyzed to estimate the effects of factors, and the analysis of variance (ANOVA) technique was employed to identify the significance of factors statistically.

### 2.6. Preparation of Fermented Juice

Fermented sea buckthorn juice was prepared according to the method proposed by Markkinen et al. [7], along with some modifications (Figure 1). The juice mixture was pasteurized at 78 °C/60 s and cooled to 35 °C for inoculation with probiotic lactic acid bacteria *L. plantarum* RM1 (MF817708). The juice was divided into 2 equal portions—control sea buckthorn juice (SBJ) and sea buckthorn juice inoculated with *Lactobacillus plantarum* cells (SBJ-Pl)—and then stored at 4 °C until further analyses.

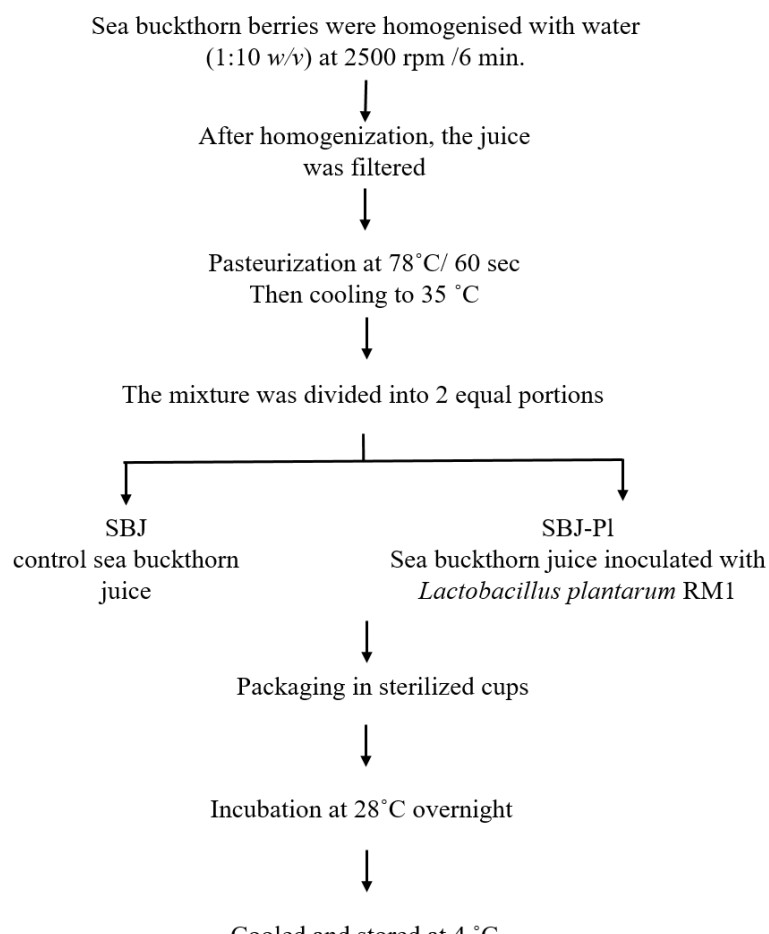

**Figure 1.** Manufacture steps of fermented sea buckthorn juice with *Lactobacillus plantarum* RM1.

*2.7. Phytochemical Content*

Total phenolic content (TPC) was determined using the Folin–Ciocalteu method and total flavonoid content (TFC) using $NaNO_2$, $AlCl_3 \cdot 6H_2O$ and NaOH according to the protocols reported by Shehata et al. [26] using a T80 UV/V is spectrometer (PG Instruments LDT, Leicestershire, United Kingdom). Gallic acid was applied as a standard for TPC, and QE as a standard for TFC. Results were expressed as (µg GAE/g dw) and as (µg QE/g dw), respectively. Total carotenoids content (TCC) was analyzed using the colorimetric method described by Dewanto et al. [27]. Briefly, 1 mL of extract (dissolved in ethanol) was added to 0.5 mL of 0.5 % NaCl, vortexed for 30 s, and centrifuged at $1500 \times g$ for 10 min. The supernatant was diluted, and the absorbance at 460 nm was measured. The amount of TCC was calculated by plotting a calibration curve with β-carotene as standard (0–0.5 mg/mL). The saponin content of the sample was determined using the vanillin-sulphuric acid assay method, reported previously by Ustundag and Mazza [28], with modification. The saponin (from plant) (Himedia Laboratories Pvt. Ltd., Maharashtra, India) was used as standard, and the total phenolic content in the juices were compared. The sulfated polysaccharides were measured by the toluidine blue assay, as described by Albano and Mourao, [29].

*2.8. Determination of Phytochemical Composition by RP-HPLC*

The phytochemical composition of sea buckthorn was assessed by the RP-HPLC technique according to Shehata et al. [30], using the VWD Agilent1260 infinity HPLC Series (Agilent, Santa Clara, CA, USA) at 284 nm wavelengths with a C18 column (aKinetex®5lJm EVO C18 106 mm × 4.6 mm, Phenomenex, Torrance, CA, USA) maintained at 35 °C [31,32]. The flow rate of the binary elution phase (A: 0.1% trifluoroacetic acid in water; B: 50% acetonitrile,

49.8% water, 0.2% trifluoroacetic acid) was 1.0 mL/min using a ternary linear elution gradient with (A) HPLC grade water 0.2 % $H_3PO_4$ (*v/v*), (B) methanol and (C) acetonitrile. Measured values were expressed as milligrams per 100 g of dry weight of berries (mg/100 g DW).

*2.9. Antioxidant Potentials*

2.9.1. The DPPH Radical Scavenging Activity

The anti-radical activities of the extracts were determined by their reaction with stable 2,2-diphenyl-1-picrylhydrazyl (DPPH) radical dissolved in absolute ethanol [33]. The reaction requires preparation of a mixture of 500 μL of extracts at various concentrations with 375 μL of ethanol and 125 μL of DPPH solution (0.02% prepared in ethanol). A control containing 875 μL of ethanol and 125 μL of DPPH solution was also prepared. After incubation for 60 min in the dark, the absorbance at 517 nm was measured. The anti-radical activity was determined using the following formula:

$$\text{The inhibition of DPPH radical \%} = \frac{\text{Abs control} - \text{Abs sample}}{\text{Abs sample}} \times 100 \tag{3}$$

Notably, a lower absorbance by the reaction mixture indicated a higher DPPH scavenging activity.

2.9.2. The ABTS Radical Cation

The free radical scavenging activity was determined by the ABTS radical cation decolorization assay, described by Re et al. [34]. ABTS was dissolved in water to a concentration of 7 μm. ABTS radical cations (ABTS+) were generated by reacting the ABTS stock solution with 2.45 μm potassium persulphate (final concentration), and the mixture was kept in the dark at room temperature for 12–16 h before use. The radical was stable in this form for more than two days when stored in the dark at room temperature. To study the radical-scavenging ability of the prepared solutions, the samples containing ABTS solution were diluted with distilled water to an absorbance of 0.700 ($\pm$ 0.02) at 734 nm and equilibrated at 30 °C. A reagent blank reading was taken (A0). After the addition of 1 mL of diluted ABTS+ solution (A734 nm = 0.700 $\pm$ 0.02) to 1 mL of extract, the absorbance was read exactly 6 min after the initial mixing (At).

$$\text{The inhibition of ABTS radical \%} = \frac{\text{Abs control} - \text{Abs sample}}{\text{Abs sample}} \times 100 \tag{4}$$

*2.10. Antimicrobial Potentials Determination*

The antimicrobial activity of the sea buckthorn berries extract was performed using the agar well diffusion method [15,35]. Ten species are known to be pathogenic to humans, including *Bacillus cereus* ATCC49064, *Candida albicans* ATCC MYA2876, *Clostridium botulinum* ATCC3584, *Escherichia coli* BA12296, *Klebsiella pneumoniae* ATCC12296, *Listeria innocua* ATCC33090, *Listeria monocytogenes* ATCC 19116, *Salmonella senftenberg* ATCC8400, *Staphylococcus aureus* NCTC10788 *and Yersinia enterocolitica* ATCC23715, which were all used in this study. One hundred microliters of the inoculum ($1 \times 10^8$ CFU/mL) were mixed with specific media for each microorganism and poured into the petri plate. One hundred microliters of the test compound were introduced into the well. The plates were then incubated overnight at 37 $\pm$ 2 °C for bacteria and 28 $\pm$ 2 °C for fungi, and the diameter (mm) of the resulting zone of inhibition was measured, including well diameter (5 mm).

*2.11. Determination of ACE Inhibitory Activity*

The ACE inhibitory activity was assayed as reported by Nakamura et al. [36]. Different concentrations of test samples in a volume of 80 μL were added to 200 μL of 5 mmol/L HHL and preincubated at 37 °C for 3 min. Test sample and HHL were prepared in 100 mmol/L borate buffer (pH 8.3) containing 300 mmol/L NaCl. The reaction was then initiated by adding 20 μL of 0.1 U/mL angiotensin-I converting enzyme from rabbit lung prepared in

100 mmol/L borate buffer (pH 8.3). After incubation at 37 °C for 30 min, the enzymatic reaction was stopped by adding 250 μL of 0.05 mol/L HCl. The liberated hippuric acid (HA) was extracted with ethyl acetate (1.7 mL) and then evaporated at 95 °C for 10 min. The residue was dissolved in 1 mL of distilled water and the absorbance of the extract at 228 nm was determined using a UV-visible spectrophotometer. ACE inhibitory activity was calculated using the equation:

$$\text{ACE inhibition } (\%) = \frac{B - A}{B - C} \times 100 \tag{5}$$

where $A$ is the absorbance of HA generated in the presence of the ACE inhibitor component, $B$ is the absorbance of HA generated without ACE inhibitors, and $C$ is the absorbance of HA generated without ACE (corresponding to HHL autolysis during enzymatic assay). The $IC_{50}$ value was defined as the concentration of inhibitor required to reduce the hippuric acid peak by 50% (indicating 50% inhibition of ACE). The method of Bradford [37], using bovine serum albumin as a standard, was used.

### 2.12. Sensory Evaluation

The organoleptic properties of the juice were evaluated using the qualitative descriptive analysis method, as previously described by Oliveira et al. [38] with slight modification. In total, 15 participants (9 males and 6 females) from the City of Scientific Research and Technological Applications (SRTA-City), Alexandria, Egypt, were asked to evaluate the desired attributes in the juice tested sample. The tasting was done in white light, and samples were presented at standard room temperature (20 ± 2). The juice samples were scored using a hedonic scale ranging from 1 to 9.

### 2.13. Cytotoxic Effect and Anticancer Activity on Human Cell Lines

Cell viability was assessed by the mitochondrial dependent reduction of yellow MTT (3-(4,5-dimethylthiazol-2-yl)-2,5-diphenyl tetrazolium bromide) to purple formazan [39]. Mouse hippocampal HT22 cells and Caco-2 Cells were suspended in DMEM medium, 1% antibiotic–antimycotic mixture (10,000 U/mL potassium penicillin, 10,000 μg/mL streptomycin sulfate and 25 μg/mL amphotericin B) and 1% L-glutamine at 37 °C under 5% $CO_2$. Cells were batch cultured for 10 days, then seeded at a concentration of $10 \times 10^3$ cells/well in fresh complete growth medium in 96-well microtiter plastic plates at 37 °C for 24 h under 5% $CO_2$ using a water jacketed carbon dioxide incubator (Sheldon, TC2323, Cornelius, OR, USA). Media were aspirated, fresh medium (without serum) was added and then cells were incubated either alone (negative control) or with different concentrations of sample to give a final concentration of (100, 50, 25, 12.5, 6.25, 3.125 and 0.78 μ g/mL). After 72 h of incubation, the medium was aspirated, and 40 μL MTT salt (2.5μg/mL) was added to each well and incubated for a further four hours at 37 °C under 5% $CO_2$. The absorbance was then measured using a microplate multi-well reader (Bio-Rad Laboratories Inc., model 3350, Hercules, CA, USA) at 595 nm, and a reference wavelength of 620 nm.

### 2.14. Statistical Analysis

The data were presented as mean values ± standard deviation. Statistical analysis was performed using one-way analysis of variance (ANOVA), followed by Duncan's test. The differences were considered significant at ($p \leq 0.05$). The IBM SPSS Statistics 23 software program was applied for statistical analyses (IBM Corp. [40], IBM SPSS Statistics for Windows, Version 23.0. IBM Corp., Armonk, NY, USA).

## 3. Results and Discussion

### 3.1. Nutritional Analysis

The percentage presence of nutrients, minerals and daily values were determined for sea buckthorn berries, and the results are illustrated in Table 1. Moisture content was the highest, at 81.90% in sea buckthorn berries. Sea buckthorn berries showed presence of fibers

in a good amount (6.55 g/100 g), covering more than 25% of daily fiber intake requirements. Berries contained only 4.45% of sugar that covers only 8.9 DV%, while total carbohydrate content including both sugars and fiber was 11 g/100 g (3.66 DV%). Fat content was 2.50 g/100 g (3.84 DV%). Sea buckthorn berries are not considered a source of protein due to its low protein content (3.12%, 6.24 DV%). Moreover, results showed 79 calories of energy per 100 g, depicting low energy production from these berries. Chemical composition of sea buckthorn berries was reported to vary due to different origin, variety and climate conditions. Obtained results are in agreement with Nazir et al. [3] and Ursache et al. [5].

**Table 1.** Chemical composition and mineral content of sea buckthorn berries.

| Parameter | Sea Buckthorn Fruits | DV% |
|---|---|---|
| Energy | 78.98 K Cal | – |
| Nutrients | g/100 g | |
| Protein | 3.12 ± 0.15 | 6.24 |
| Fat | 2.50 ± 0.32 | 3.84 |
| Fiber | 6.55 ± 0.28 | 26.20 |
| Total sugars | 4.45 ± 1.30 | 8.9 |
| Total carbohydrates | 11 | 3.66 |
| Moisture | 81.90 ± 1.30 | – |
| Ash | 1.47 ± 0.27 | – |
| Minerals | mg/100 g | |
| Phosphorus (P) | 110.72 | 11.07 |
| Sodium (Na) | 155.31 | 6.47 |
| Manganese (Mn) | 0.98 | 0.24 |
| Potassium (K) | 793.43 | 22.66 |
| Copper (Cu) | 0.43 | 21.81 |
| Gallium (Ga) | 1.90 | – |

Data represented are the mean ± SD. Energy, total carbohydrates and % DV are calculated based on means values ± SD. DV represents the daily values for nutrition labelling, calculated based on a caloric intake of 2000 calories, for adults and children aged 4 years or more.

The evaluation of mineral content in sea buckthorn showed that potassium is the most abundant of all the elements investigated in the berries (793.43 mg/100 g), and can achieve 22.66 of its DV%. It is well known that potassium plays an important role in the ionic balance and helps in maintaining the tissue excitability and cell function of the human body [41]. The content of copper and phosphorus were 21.81 and 11.07 DV%, respectively. Despite the high sodium content (155.31 mg/100 g), it only represented 6.47 DV%. On the other hand, sea buckthorn contained trace amounts of manganese and gallium (0.98, 1.90 mg/100 g). Bal et al. [42] reported varied mineral content of sea buckthorn according to origin and plant parts.

Amino acid composition and amino acid scores (AAS) of sea buckthorn berries are represented in Table 2. Although there was a limited content of protein in sea buckthorn berries (3.12 g/100 g), it showed a unique balance of essential amino acid content. Histidine showed the highest AAS (24.32%), with a content of 3.89 mg/g of daily requirements, followed by valine, threonine, leucine and lysine with AAS 23.66, 23.09, 23.05 and 21.71% and content of 3.08, 2.08, 4.38 and 3.47 mg/g, respectively. The limiting amino acids were isoleucine and phenylalanine with AAS 15.79 and 16.95% and content of 2.05 and 3.22 mg/g, respectively, and there was an absence of methionine and cystine. Concerning non-essential amino acids, sea buckthorn berries showed remarkable amino acid content that exceeded 100% of daily requirements of glycine, alanine and aspartic acid (157.15, 128.15 and 106.23%, respectively). Sea buckthorn provided approximately half of the daily requirements of glutamic acid, serine, proline and tyrosine (62.17, 58.08, 43.33 and 42.50%, respectively) and an abundant amount of arginine (5.86 mg/g) (the AAS of this amino acid was not documented). The amino acid composition and score are reflected in the high quality of sea buckthorn protein. These results are in agreement with Ciesarová et al. [43], who stated that the full range of proteinogenic amino acids and essential amino acids

content in sea buckthorn is high, and as such it is within a class of relatively high-quality plant protein resources.

**Table 2.** Amino acid content of sea buckthorn.

| Amino Acid | Symbol | mg/g | AAS |
|---|---|---|---|
| *Essential amino acids* | | | |
| Histidine | His | 3.89 | 24.32 |
| Leucine | Leu | 4.38 | 23.05 |
| Isoleucine * | Ile | 2.05 | 15.79 |
| Lysine | Lys | 3.47 | 21.71 |
| Methionine + cystine | Met + Cys | ND | ND |
| Phenylalanine * | Phe | 3.22 | 16.95 |
| Threonine | Thr | 2.08 | 23.09 |
| Valine | Val | 3.08 | 23.66 |
| *Non-essential amino acids* | | | |
| Alanine | Ala | 3.33 | 128.15 |
| Aspartic acid | Asp | 9.35 | 106.23 |
| Glutamic acid | Glu | 10.88 | 62.17 |
| Glycine | Gly | 3.14 | 157.15 |
| Proline | Pro | 2.64 | 43.33 |
| Serine | Ser | 3.08 | 58.08 |
| Tyrosine | Tyr | 1.96 | 42.50 |
| Arginine | Arg | 5.86 | — |

Pattern (mg/g protein) for adults according to (FAO/WHO/UNU, 1985), based on highest estimate of requirement to achieve nitrogen balance (estimated amino acid requirements in adults) (FAO/WHO, 1973), assuming a safe level of protein intake of 0.55 g per kg per day (averaged value for men and women) (Recommended Dietary Allowances, Food and Nutrition Board, Commission on Life Sciences, National Research Council, 1989). AAS, amino acid score. * Limiting amino acids.

### 3.2. Optimization of Variables of Probiotic Growth Using CCD-RSM

Central composite design (CCD) was used to determine the optimum levels and factors interactions (Table S1). Juice concentration, inoculum size and temperature were selected for optimization by RSM. Table S2 represents the effect of the three independent variables on bacteriocin production yield determined by CCD through 20 runs. The maximum growth of *L. plantarum* obtained from RSM was 8.5 (log CFU/mL). According to Equation (3), the regression analysis data were fitted to a quadratic model, and the second order regression equation obtained was the full actual model on bacteriocin production.

$$Y \text{ (Probiotic Growth)} = \beta_0 + \beta_1 X_1 + \beta_2 X_2 + \beta_3 X_3 + \beta_{11} X_{12} + \beta_{22} X_{22} + \beta_{33} X_{32} + \beta_{12} X_1 X_2 + \beta_{13} X_1 X_3 + \beta_{23} X_2 X_3 \tag{6}$$

where $Y$ is probiotic growth (log cfu/mL), $A$ is juice concentration (g/100 mL), $B$ is inoculum size (mL/100 mL) and $C$ is temperature (°C). The optimum condition that produced the highest of probiotic growth was juice concentration (20 g/100 mL), inoculum size (1 mL/100 mL), and with a temperature of 37 °C incubation for 20 h (Run 20). The value of predicted $R^2$ (0.9838) and adjusted $R^2$ (0.9677) were almost equal. Moreover, a high value of coefficient of determination ($R^2$ = 0.9838) revealed appropriate adjustment of the mathematical model. This $R^2$ value showed that the model could explain 99.92% of variations in response to the independent variables [44]. The model F-value (60.89) (Figure 2) implied the significance of the model for probiotic growth by the coefficient of determination ($R^2$), i.e., 0.9893, reflecting a good agreement between the experimental and the predicted responses. The value of "p > F" less than 0.05 indicated that the model terms were also significant. Furthermore, the linear term A, B and C, and quadratic terms A2, B2 and C2 were also found to be significant. Consequently, the current research showed that juice concentration, inoculum size and temperature were the main key factors in probiotic growth, with a 95% confidence level.

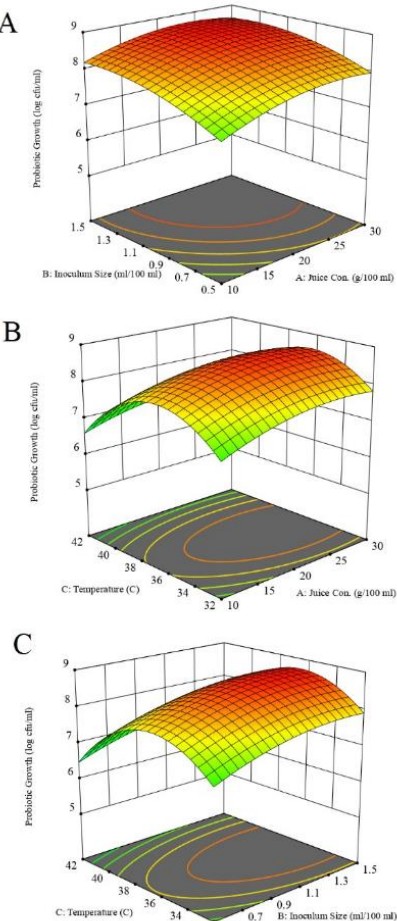

**Figure 2.** 3D response surface plot presenting the effect of interactions of independent variables on *Lactobacillus plantarum* growth. (**A**) Juice concentration vs. inoculum size; (**B**) temperature vs. juice concentration; (**C**) temperature vs. inoculum size.

The 3D response surface plots (Figure 2) were illustrated to understand interactive effects of each fermentation parameter (variable) on probiotic growth. The contour plots illustrate the modulatory effect of temperature, inoculum size and juice concentration on probiotic growth. The results suggest interaction between abiotic parameters in regulating the growth of probiotic bacteria.

In Figure 3, normal probability vs. residuals was plotted, and showed that the data were very similar to the straight line and positioned on both sides, suggesting the reasonability of the model. The difference between the real and predicted values is residuals [45]. The standard percent probability vs. studentized residuals showed that the precise values provide the sufficient estimated model. Moreover, the residual plot along a straightaway line was satisfactory with a normality assumption, which proved the accuracy of CCD.

Validation of the Optimum Condition

The experiment was performed in duplicates, with an optimized probiotic growth in sea buckthorn juice containing juice concentration 20 g/100 mL, inoculum size 1 mL/100 mL, and with a temperature of 37 °C incubation for 20 h to evaluate the optimization results and establish the accuracy of the model. Upon experimentation, the observation of probiotic growth reaching 8.5 log cfu/mL proves the high suitability of the model.

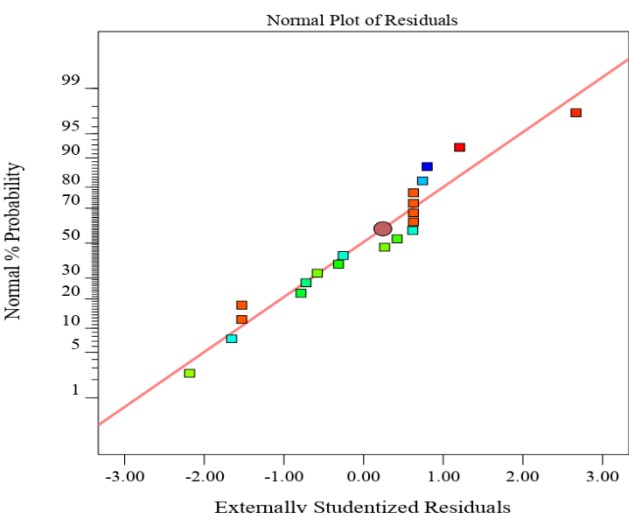

**Figure 3.** Normal probability plot of the residuals of a quadratic polynomial model for *Lactobacillus plantarum* growth.

### 3.3. Phytochemical and Physical Analysis

Phytochemical screening of sea buckthorn juice is shown in Table 3. The analysis indicated that sea buckthorn juice (SBJ) is rich in phenolic, flavonoid and carotenoid content (283.58, 118.42 and 6.5 mg/g, respectively), in addition to its high content of vitamin C (322.33 mg/g), in parallel with trace content of sulphate polysaccharides (0.31 mg/g). The obtained results showed the effectiveness and importance of the fermentation technology in the improving of the quality of the buckthorn juice, with the fermentation of the juice with *Lactobacillus plantarum* resulting in a significant increase in the phenolic and flavonoid content ($p \leq 0.05$) (Table 3). Total phenolic content increased from $283.58 \pm 1.45$ mg/g in the nonfermented juice to $304.16 \pm 1.31$ mg/g in the fermented sample ($p \leq 0.05$). Here, it should be mentioned that the existence of vitamin C may interfere in the measuring of the total phenolic and flavonoid content due to raising their content in the juice; however, in all cases, an increase of antioxidant capacity of the fermented juice was observed. On the other hand, the fermentation process did not show any significant effect in the content of saponins, carotenoids and vitamin C. The results are in agreement with Aaby et al. [46]. Variability of phenolic contents has been recorded in sea buckthorn berry juice due to the influence of a number of factors, such as the type of locality and the ripening time [47]. The content of phenolics, flavonoids and saponin in in the sea buckthorn juice before and after fermentation is pivotal for the final therapeutic properties of the juice post-fermenta-tion. The role of phenolics and saponin in sea buckthorn have been explored for their anticoagulant potential; the prevention of oxidative damage to DNA [48], as well as the prevention of lipid peroxidation and protein carbonylation have been recently reported [49]. The insignificant difference in the saponin of the juice sample may be pivotal in the nutritional and therapeutic relevance of the juice, post-fermentation.

The physical properties of sea buckthorn juice reflected its acidity with a pH of 3.01, a TA of 2.55 g/100 g and a solid content of 8.25 °Bx. These results were in agreement with Araya-Farias et al. [14].

The results of sea buckthorn juice fermented with *Lactobacillus plantarum* (SBJ-Pl) indicated a significant increase in phenolics, flavonoids and vitamin C content ($304.16 \pm 1.31$, $131.2 \pm 1.68$ and $327.4 \pm 1.70$ mg/g, respectively), as well as trace content of sulphate polysaccharides ($0.42 \pm 0.042$ mg/g). Similar observation of the lactic acid fermentation effect on sea buckthorn berry phenolic compounds was recorded by Markkinen et al. [7], who advised careful selection of strains and optimization of the process in order to maintain the contents and compositions of health-promoting phenolic compounds during fermentation as a strain dependent process.

Physical analysis of both sea buckthorn fermented juice and SBJ-Pl showed a slight lowering in pH up to (2.77 ± 0.095) and insignificant acidity (2.96 ± 0.123 g/100 g), with insignificant differences in solid content of 8.34 °Bx, respectively. These results agreed with Araya-Farias et al. [14].

**Table 3.** Phytochemical and physical analysis of sea buckthorn juice (mg/g).

| Parameter | Unit | SBJ | SBJ-Pl |
|---|---|---|---|
| Total phenolics | mg/g | 283.58 ± 1.45 [c] | 304.16 ± 1.31 [a] |
| Total flavonoids | mg/g | 118.42 ± 1.82 [c] | 131.2 ± 1.68 [a] |
| Total carotenoids | mg/g | 6.5 ± 0.24 [a] | 6.83 ± 0.28 [a] |
| Vitamin C | mg/g | 322.33 ± 2.05 [c] | 327.4 ± 1.70 [a] |
| Total saponin | mg/g | 13.71 ± 0.17 [a] | 13.74 ± 0.09 [a] |
| Sulphate polysaccharide | μg/g | 0.31 ± 0.029 [b] | 0.42 ± 0.042 [ab] |
| pH | – | 3.01 ± 0.094 [a] | 2.77 ± 0.095 [ab] |
| Soluble solids | °Bx | 8.25 ± 0.18 [a] | 8.34 ± 0.14 [a] |
| Titratable acidity | g citric acid/100 g | 2.55 ± 0.066 [b] | 2.96 ± 0.123 [b] |

Data represented in means of duplicates ± SD. Total phenolics were expressed as gallic acid equivalents (GAE), mg/g sample. Total flavonoids were expressed as Quercetin equivalent (QE), mg/g sample. Total saponin was expressed as mg/g sample. Mean values in a line with different superscripts letters are significantly different at $p \leq 0.05$. SBJ, sea buckthorn juice; SBJ-Pl, sea buckthorn juice fermented with *Lactobacillus plantarum*.

### 3.4. HPLC Analysis of Phenolics and Flavonoids

HPLC analysis of phenolic and flavonoid compounds of fermented and non-fermented sea buckthorn juice is represented in Table 4. Benzoic acid is the dominant phenolic compound in non-fermented sea buckthorn berries (3825.90 mg/kg), which showed a significant increase up to 6288.21 mg/kg in fermented juice ($p \leq 0.05$). Benzoic acid has a crucial role as an antimicrobial agent in food preservation [50], followed by abundant contents of resveratrol, rutin, p- hydroxy benzoic acid, kampherol, gallic acid, quinol and ellagic acid (435.06, 422.30, 262.10, 244.50, 178.20, 127.68 and 94.70 mg/kg, respectively), which were significantly increased in fermented juice ($p \leq 0.05$) (Table 4). Phenolic and flavonoid compounds quercitrin, rosemarinic acid, caffeic acid, p-coumaric acid, o- coumaric acid, syringic acid and chlorogenic acid showed average contents with concentrations of 59.63, 55.91, 40.55, 40.15, 40.13, 27.57 and 24.20 mg/kg, respectively. Sea buckthorn juice showed poor content of cinnamic acid and catechin (9.01, 8.02 mg/kg, respectively). Similar results were reported by Bittová et al. [51], who demonstrated that the total content of sea buckthorn polyphenolic compounds was more announced in fruits. Fermentation of sea buckthorn juice with *Lactobacillus plantarum* led to a significant increase in most of phenolic compounds, except for ellagic acid and quercetin, which decreased from 94.70 and 59.63 mg/kg in non-fermented juice to 22.28 and 30.94 mg/kg in the fermented sample, respectively. Quinol was totally converted as it was not detected in fermented juice; on contrary, myricetin (1007.67 mg/kg) and ferulic acid (13.57 mg/kg) were detected in fermented juice SBJ-Pl. Natural antioxidants, such as phenolic compounds, are mainly present in conjugated and bound forms (conjugates with sugars, fatty acids or proteins) in the non-fermented form. Fermentation was reported to be a suitable technique to enhance the release of bound phenolic compounds before consumption, as it increases the bioconversion of phenolic compounds from their linked or conjugated forms to their free ones, resulting in an increase in their concentration with greater antioxidant power [52–54]. The obtained results are in agreement with Markkinen et al. [7]. On the other hand, Negi and Dey [55] reported a slight decrease in phenolic compounds under fermentation conditions.

**Table 4.** HPLC analysis of the phenolic, flavonoid and organic acid content of sea buckthorn juice.

| Phenolics/Flavonoids | Conc. (mg/kg) | |
|---|---|---|
| | **SBJ** | **SBJ-Pl** |
| Pyrogallol | ND | ND |
| Quinol | 127.68 | ND |
| Gallic acid | 178.20 | 325.45 |
| 3-Hydroxytyrosol | ND | ND |
| Catechol | ND | ND |
| p- Hydroxy benzoic acid | 262.10 | 352.03 |
| Catechin | 8.02 | 12.64 |
| Chlorogenic acid | 24.20 | 50.57 |
| Vanillic acid | ND | ND |
| Caffeic acid | 40.55 | 66.47 |
| Syringic acid | 27.57 | 37.45 |
| p- Coumaric acid | 40.15 | 59.49 |
| Benzoic acid | 3825.90 | 6288.21 |
| Ferulic acid | ND | 13.57 |
| Rutin | 422.30 | 664.41 |
| Ellagic acid | 94.70 | 22.28 |
| O- Coumaric acid | 40.13 | 341.10 |
| Resveratrol | 435.06 | 705.47 |
| Cinnamic acid | 9.01 | 20.55 |
| Quercitin | 59.63 | 30.94 |
| Rosemarinic acid | 55.91 | 56.79 |
| Neringein | ND | ND |
| Myricetin | ND | 1007.67 |
| Kampherol | 244.50 | 328.42 |

Concentrations are expressed as (mg/kg). SBJ, sea buckthorn juice; SBJ-Pl, sea buckthorn juice fermented with *Lactobacillus plantarum*.

### 3.5. Antioxidant Activity

Figure 4A,B illustrated the antioxidant power of non-fermented (SBJ) and fermented (SBJ-Pl) sea buckthorn juice in comparison with the application of ascorbic acid in both DPPH and ABTS methods, respectively. Both non-fermented (SBJ) and fermented (SBJ-Pl) juices showed significant antioxidant activity, which were higher than ascorbic acid (as standard antioxidant) in both DPPH and ABTS methods (Figure 4A,B, respectively) ($p \leq 0.05$) and were directly proportional to concentration. The high antioxidant power in sea buckthorn juice might be mainly due to its rich of phenolic and flavonoid content (Tables 3 and 4). These results are consistent with the reported antioxidant potentials of sea buckthorn of previous research [6,56,57]. Fermentation of sea buckthorn juice with *Lactobacillus plantarum* (SBJ-Pl) led to a significant increase in its antioxidant potentials either by DPPH (Figure 4A) or ABTS (Figure 4B). This increase in antioxidant activity is related to the increase in most of the phenolic compounds in fermented juice (revealed in Table 4), which agreed with the results reported by Markkinen et al. [7].

### 3.6. Antimicrobial Activity

The antimicrobial activity of non-fermented (SBJ) and fermented (SBJ-Pl) sea buckthorn juice against 10 food-borne pathogens is presented in Table 5. The results of the current research showed anti-microbial power of both non-fermented and fermented juice, with a significant increase in fermented juice ($p \leq 0.05$). The sea buckthorn juice showed remarkable antimicrobial potentials against all tested pathogens; the highest effect was noted against *Escherichia coli* BA12296, with an inhibition zone of 17.46 ± 0.41 mm for SBJ and 22.33 ± 1.64 mm for SBJ-Pl. On the other hand, the least effect was remarked against *Clostridium botulinum* ATCC3584, with an inhibition zone of 8.83 ± 1.64 mm for SBJ and 11.5 ± 1.08 mm for SBJ-Pl. The high antimicrobial potentials of sea buckthorn might be due to its rich content of phenols and flavonoids (Tables 3 and 4), which facilitates its

recommendation as a food preservative [50] in food applications. Lactic acid fermentation of sea buckthorn juice SBJ-Pl significantly increased its antimicrobial potential against *Escherichia coli* BA12296 (22.33 mm), *Klebsiella pneumoniae* ATCC12296 (13.76 mm) and *Listeria innocua* ATCC33090 (14.5 mm) due to the increase of the acidity of the fermented juice beside its phenolic content. These results are in accordance with previous results (Tables 3 and 4) when fermentation significantly increased phenolic content and enhanced their potentials. Many researchers have previously reported the antimicrobial potentials of sea buckthorn berries [58–60].

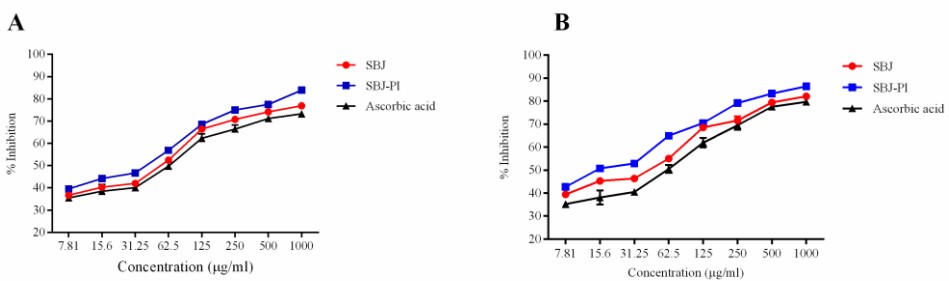

**Figure 4.** Antioxidant activity of fermented and non-fermented sea buckthorn juice. (**A**) DPPH methods; (**B**) ABTS methods. SBJ, sea buckthorn juice; SBJ-Pl, sea buckthorn juice fermented with *Lactobacillus plantarum*.

**Table 5.** Antimicrobial activity of sea buckthorn juice.

| Buckthorn Juice / Pathogenic Microorganisms | SBJ IZD (mm) | SBJ-Pl IZD (mm) |
|---|---|---|
| *Bacillus cereus* ATCC49064 | 12.4 ± 0.88 [a] | 14.4 ± 0.75 [a] |
| *Candida albicans* ATCCMYA2876 | 10.6 ± 1.96 [a] | 13.33 ± 0.62 [a] |
| *Clostridium botulinum* ATCC3584 | 8.83 ± 1.64 [a] | 11.5 ± 1.08 [a] |
| *Escherichia coli* BA12296 | 17.46 ± 0.41 [b] | 22.33 ± 1.64 [a] |
| *Klebsiella pneumoniae* ATCC12296 | 9.66 ± 1.02 [b] | 13.76 ± 1.32 [a] |
| *Listeria innocua* ATCC33090 | 10.5 ±1.47 [b] | 14.5 ± 0.75 [a] |
| *Listeria monocytogenes* ATCC19116 | 9.13 ± 1.18 [b] | 13.96 ± 1.03 [a] |
| *Salmonella senftenberg* ATCC8400 | 15.13 ± 0.98 [b] | 21.3 ± 0.61 [a] |
| *Staphylococcus aureus* NCTC10788 | 16.26 ± 1.11 [b] | 21.3 ±0.66 [a] |
| *Yersinia enterocolitica* ATCC23715 | 14.33 ± 0.49 [b] | 20.8 ± 1.66 [a] |

- Data represented are means of duplicates of inhibition zone diameter (IZD) (mm) ± SD. - Mean values in a line having different superscript are significantly different at ($p \leq 0.05$). - SBJ, Sea buckthorn juice; SBJ-Pl, Sea buckthorn juice fermented with *Lactobacillus plantarum*.

### 3.7. Angiotensin-Converting Enzyme (ACE) Inhibitory Activity

ACE inhibitors are vastly used for the remediation of cardiovascular diseases by relaxing the veins and arteries, thus lowering the blood pressure. Sea buckthorn juice samples (250–1000 µg/mL) were tested for their ACE inhibitory activity (Figure 5A) and $IC_{50}$ value (Figure 5B). Sea buckthorn juice fermented with *Lactobacillus plantarum* (SBJ-Pl) exhibited higher ACE inhibitory activity (76.51 ± 2.76%) than non-fermented sea buckthorn juice (SBJ) (56.10 ± 1.90%) at a 1000 µg/mL concentration. ACE inhibition of sea buckthorn juice at 205 and 500 µg/mL concentration showed a similar pattern to juice at 1000 µg/mL concentration. The $IC_{50}$ value is defined as the concentration of sea buckthorn juice required to inhibit 50% of the ACE activity. The juice fermented with *Lactobacillus plantarum* exhibited a high ACE inhibitory activity, showing the lowest $IC_{50}$ value of 222.87 ± 3.59 µg/mL. The ACE inhibitory activity of phenolic extracts from plants have been reported in previous studies [61,62]. Phenolic compounds have an ability to modify the structure of ACE enzyme and reduce its activity by interacting with the disulphide bridges presented on the surface of the enzyme [63]. Therefore, inhibition of ACE by phenolics of the sea buckthorn juice can be considered as an unusual and beneficial treatment of hypertension. The findings of the

current study of selected sea buckthorn juice agreed with earlier reports on the inhibition of ACE by phenolic extracts of bitter leaf [64], soybean [65] and *Allium sativum* from garlic [66].

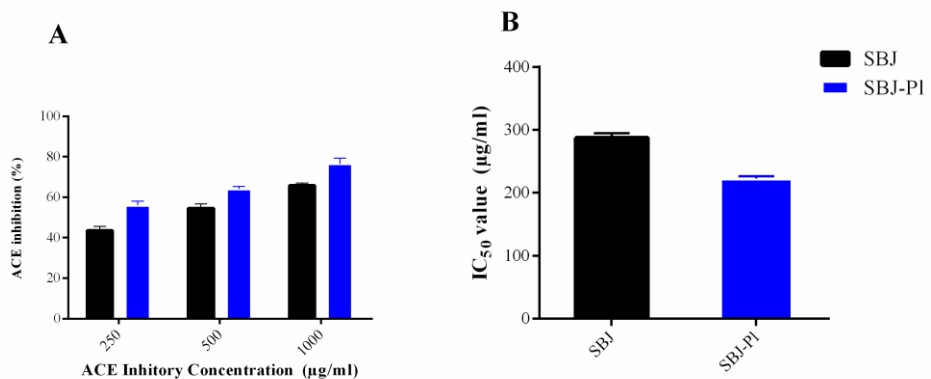

**Figure 5.** ACE inhibitory activity of sea buckthorn juice extracts. (**A**) ACE inhibitory activity of sea buckthorn juice types; (**B**) $IC_{50}$ (mg/mL) value of sea buckthorn juice types. SBJ, sea buckthorn juice; SBJ-Pl, sea buckthorn juice fermented with *Lactobacillus plantarum*.

*3.8. In Vitro Cytotoxicity and Anticancer Activity of Sea Buckthorn Juice*

The safety of sea buckthorn juice extract has been tested on normal human lung fibroblast cells (WI-38 cell line), represented by the $EC_{100}$ values. The results emphasized that buckthorn juice extract is safe in a dose up to $418.11 \pm 0.86$ μg/mL. The safety of an anticancer drug is one of the most critical concerns in the treatment of cancer patients. The anticancer activity expressed as $IC_{50}$ (concentration of the studied extracts that caused 50% cell death) is usually used to represent the strength and sensitivity of the drug on cancer cell lines. Figure 6 demonstrated that the extract of sea buckthorn fermented juice (SBJ-Pl) had the highest anticancer effect against the two studied cancer cell lines (Caco-2 and HT-22). Moreover, there was a significant difference recorded between the $IC_{50}$ values of the extract from non-fermented sea buckthorn juice (SBJ) and fermented sea buckthorn juice (SBJ-Pl) with *L. plantarum* for the two cancer cell lines [67,68]. Various studies by Dienaitė [69] and Guo [59] reported the anticancer potential activity of sea buckthorn against HT-29 and Caco-2 cell lines. Sea buckthorn extracts and its polyphenols may be potentiating their anticancer activity. Polyphenols from diet have previously been documented to interfere with many of the biochemical pathways involved in cancer progression [70]. Furthermore, polyphenols may act as an immune system booster and help protect living cells from ROS damage. Clinical trials of polyphenol treatments confirming the protective mechanism have investigated variations in dose, timing and other conflicting factors [71]. Thus, this in vitro study aimed to evaluate the anticancer potential of sea buckthorn fruit juice in non-fermented and fermented juice by lactic acid bacteria as a first step for potential use in health-promoting foods.

*3.9. Sensory Evaluation of Sea Buckthorn Juice*

The sensory properties of sea buckhorn juice (Figure 7) showed that, as the titratable acidity as citric acid changed from 2.55 to 2.96 g/100 g when fermented, the fermented juice scored a little bit less than non-fermented mainly due to a slight increase in sourness (as described by panelists) that relates to its pH and titratable acidity values (Table 3). Consequently, no significant differences in the titratable acidity have been reported between fermented and non-fermented juice, which emphasized the insignificant sourness effect of the consumers acceptability. Influences of various factors including chemical composition and phenolic compounds on sensory perception are also reported [43].

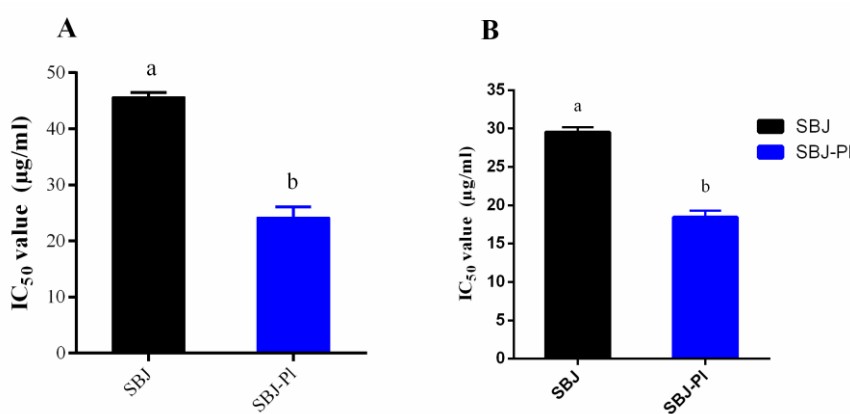

**Figure 6.** IC$_{50}$ (μg/mL) of different extracts of normal and fermented sea buckthorn juices against proliferation of human cancer cell lines: (**A**) Caco-2 cell line; (**B**) HT-22 cell line. SBJ, sea buckthorn juice; SBJ-Pl, sea buckthorn juice fermented with *Lactobacillus plantarum*. Results are presented as mean $\pm$ SE (*n* = 3). Different letters in the same row are significantly different at *p* < 0.05. IC$_{50}$, concentration of the studied extracts that caused 50% viability for tested.

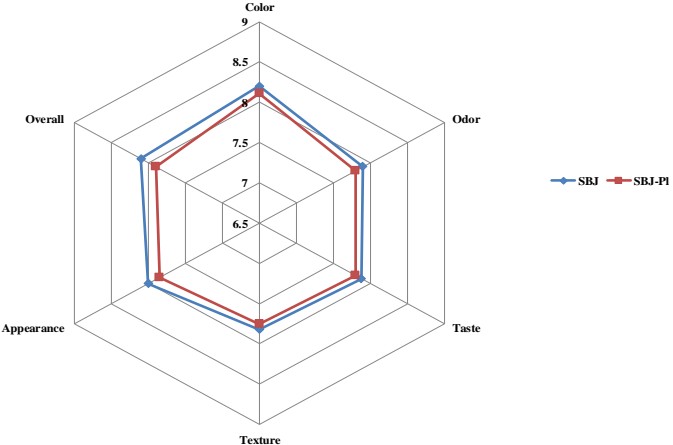

**Figure 7.** Spider plot of the sensory analysis of sea buckthorn juice. SBJ, sea buckthorn juice; SBJ-Pl, sea buckthorn juice fermented with *Lactobacillus plantarum*.

## 4. Conclusions

Sea buckthorn berries showed a considerable nutritional value, especially in fiber, minerals and high-quality protein containing essential amino acids, in parallel with a lowcalorie intake. The current study has provided an insight on the fermentation of sea buckthorn with probiotic bacteria *L. plantarum*, along with assessment of nutritional, therapeutic and sensory evaluation. The results have demonstrated the significant role of abiotic factors, such as juice concentration, inoculum size and temperature on probiotic growth, suggesting a strategic approach to develop a non-dairy probiotic beverage. The phytochemical assessments have shown an increase in the total phenolic content of the juice after fermentation, without much change in other phytoconstituents. The sea buckthorn juice pre- and post-fermentation indicated its rich phenolic, flavonoid, carotenoid and vitamin C content with good antioxidant, antimicrobial, ACE inhibitory, and anticancer potentials that significantly increased via fermentation with *L. plantarum*, in addition to consumers' sensorial acceptance. Furthermore, the findings in the current work may provide guided data for researchers regarding optimal growth levels for *L. plantarum* RM1 in sea buckthorn juice by applying response surface methodology (RSM). The fermentation of sea buckthorn juice with *L. plantarum* caused a significant increase in phenolic and flavonoid content, as well as antioxidant and antimicrobial activities. The outcome could be applied on the commercial scale for production of sea buckthorn juice to develop a

health-promoting and tasty fermented product. Based on the results, it can be interpreted that a non-dairy probiotic beverage could be a future safe alternative to dairy products. In addition, fermented juices can be assessed for the mechanism of therapeutic activity towards other diseases impacting the society at large.

**Supplementary Materials:** The following supporting information can be downloaded at: https: //www.mdpi.com/article/10.3390/fermentation8080391/s1, Table S1: experimental design and results of central composite design for optimization of pro-biotic growth in sea buckthorn juice; Table S2: experimental design and results of central composite design for optimization of pro-biotic growth in sea buckthorn juice.

**Author Contributions:** Conceptualization, M.G.S., S.A.E.-S. and A.M.; methodology, M.G.S., S.A.E.-S., N.M.A.E.-A., A.G.D. and P.U.; software, M.G.S. and N.M.A.E.-A.; validation, S.A.E.-S. and A.M.; formal analysis, M.G.S., N.M.A.E.-A. and A.G.D.; investigation, M.G.S., A.M., N.M.A.E.-A. and A.G.D.; resources, S.A.E.-S., M.G.S. and A.M.; data curation, M.G.S., S.A.E.-S. and A.M.; writing—original draft preparation, M.G.S., N.M.A.E.-A. and A.G.D.; writing—review and editing, S.A.E.-S. and P.G.; visualization, S.A.E.-S., M.G.S. and A.M.; supervision, S.A.E.-S., M.G.S. and P.G.; project administration, S.A.E.-S., M.G.S. and A.M.; funding acquisition, S.A.E.-S., M.G.S. and A.M. All authors have read and agreed to the published version of the manuscript.

**Funding:** This research was funded by the Academy of Scientific Research and Technology (ASRT), within the framework of The Bilateral Scientific and Technological Cooperation, between The Arab Republic of Egypt Ministry of Scientific Research (MoSR) and the Department of Science and Technology, Government of India, grant number DST/INT/Egypt/P-06/2019.

**Institutional Review Board Statement:** Not applicable.

**Data Availability Statement:** Not applicable.

**Acknowledgments:** This work is the part of the project supported financially by the Academy of Scientific Research and Technology (ASRT) within the framework of The Bilateral Scientific and Technological Cooperation, between The Arab Republic of Egypt Ministry of Scientific Research (MoSR) and the Department of Science and Technology, Government of India (DST/INT/Egypt/P-06/2019).

**Conflicts of Interest:** The authors declare no conflict of interest. The funders had no role in the design of the study; in the collection, analyses, or interpretation of data; in the writing of the manuscript; or in the decision to publish the results.

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
