# Peer review of "Nutritional Evaluation of Sea Buckthorn “Hippophae rhamnoides” Berries and the Pharmaceutical Potential of the Fermented Juice"

_fermentation, doi:10.3390/fermentation8080391_

Round 1
Reviewer 1 Report
Please revise the title to match the objective statement and conclusion. the conclusion is weak and requires major revision, what are the limitations and future work.
Author Response
Response to Reviewer 1 Comments
- Please revise the title to match the objective statement and conclusion.
Response: The Title of the manuscript was modified to be “Nutritional evaluation of Sea Buckthorn “Hippophae goniocarpa” berries and the pharmaceutical potential of the fermented juice”
- The conclusion is weak and requires major revision; What are the limitations and future work?
Response: The conclusion has been improved as suggested and future works have been added

Reviewer 2 Report
In this work, the authors study The Impact of fermentation of sea buckthorn juice with L. plantarum on antioxidant, antimicrobial, antihypertension and anticancer activity. The topic is interesting and the manuscript has a potential from a scientific point of view. However, there are a few points that require the authors’ attention.
1. Line 55: The meaning “genus the Lactobacillus sp.” is not clear, please edit.
2. Introduction: The second half of the Introduction section seems to be out of place. How is Sea buckthorn or fermented products connected to ACE/ACE inhibitors? More information should be added in the Introduction in order to connect the two parts.
3. Lines 68-72: At this point the aim of the study should be clearly stated. The reader should not struggle to understand the aim of the study. Try to be precise about the objectives. What was the purpose? Why are you investigating this matter?
4. Line 101: Did the authors perform freeze-drying on raw sea buckthorns? The conditions and other relevant information should be added.
5. Line 102: Please add the city and the country of origin for the equipment used. The same applies for the whole manuscript.
6. Lines 192-195: Add the origin of all the strains used.
7. Line 442: I believe you mean: “Antioxidant activity of fermented and non-fermented sea buckthorn juice”.
8. Line 533: According to section 3.10, fermentation of sea buckhorn juice resulted in lower scores for all attributes, even consumer acceptance. So, I believe this line has to be edited. If the decrease in acceptance was not statistically significant, then make it clear that the new products were still accepted by the panel despite the sour notes in taste. If however acceptance was significantly decreased, then the authors should propose a way to increase consumer acceptance. The same applies for the abstract section.
Author Response
Response to Reviewer 2 Comments
Comments and Suggestions for Authors
In this work, the authors study The Impact of fermentation of sea buckthorn juice with L. plantarum on antioxidant, antimicrobial, antihypertension and anticancer activity. The topic is interesting, and the manuscript has a potential from a scientific point of view. However, there are a few points that require the authors’ attention.
Thanks for respected reviewer for his kind effort and time to improve the manuscript
- Line 55: The meaning “genus the Lactobacillus sp.” is not clear, please edit.
Response: Modifications have been made accordingly.
- Introduction: The second half of the Introduction section seems to be out of place. How is Sea buckthorn or fermented products connected to ACE/ACE inhibitors? More information should be added in the Introduction in order to connect the two parts.
Response: Modifications have been made accordingly and highlighted with yellow color
- Lines 68-72: At this point the aim of the study should be clearly stated. The reader should not struggle to understand the aim of the study. Try to be precise about the objectives. What was the purpose? Why are you investigating this matter?
Response: The aim of the study and objectives has been reformatted to be clearer and more understandable.
- Line 101: Did the authors perform freeze-drying on raw sea buckthorns? The conditions and other relevant information should be added.
Response: Lyophilized Sea buckthorn berries were purchased from local Indian market by Amazon website in October (2020) and all storage conditions and information are presented in materials section.
- Line 102: Please add the city and the country of origin for the equipment used. The same applies for the whole manuscript.
Response: The city and country of all used scientific equipments were presented in the section of materials and methods accordingly.
- Lines 192-195: Add the origin of all the strains used.
Response: Lactobacillus plantarum RM1 (MF817708) was isolated and identified by us in the Department of Food Technology, City of Scientific Research from fermented milk (Rayeb milk) that was purchased from a local producer in Alexandria, Egypt. The strain was registered in gene bank and coded with ((MF817708).
- Line 442: I believe you mean: “Antioxidant activity of fermented and non-fermented sea buckthorn juice”.
Response: Modifications have been made accordingly.
- Line 533: According to section 3.10, fermentation of sea buckhorn juice resulted in lower scores for all attributes, even consumer acceptance. So, I believe this line has to be edited. If the decrease in acceptance was not statistically significant, then make it clear that the new products were still accepted by the panel despite the sour notes in taste. If however acceptance was significantly decreased, then the authors should propose a way to increase consumer acceptance. The same applies for the abstract section.
Response: The section has been modified to be more clear
“The sensory properties of sea buckhorn juice (Fig. 7) showed that, as the titratable acidity as citric acid changed from 2.55 to 2.96 g/100g when fermented, the fermented juice scored a little bit less than non-fermented mainly due to it’s a slight increase in sourness (as described by panelists) that relates to its pH and titratable acidity values (Table 3). Consequently, there are no significant differences in the titratable acidity has been reported between fermented and non-fermented juice, which emphasized the insignificant sourness effect of the consumers acceptability”

Reviewer 3 Report
The present paper describes the Impact of fermentation of sea buckthorn juice with L. plantarum on antioxidant, antimicrobial, antihypertension and anticancer activity
considering the aforementioned title and the lines 68-72 the reader does not expect to find in the results and discussion section the parts 3.1 and 3.2 which refer to the berries. Thus, it is redundund and out of the scope of the work. To support that it is the characterization of the raw material, other commmplementary analyses as those mmade in the juice would be required.
A further observation regarding the particular part is that the authors separate the amino acid content from chemical composition(?), whereas in the chemical composition section they put together nutrients and minerals. All the section should have been combined in one paragraph as Nutritional analysis (and within the text to coment on non-essential aino acids)
The authors comment also on % of daily requirements. Maybe I misunderstood but is there somewhere a daly intake of berries or what is considered as a portion for consummption?
2.7 methodology
Regarding the total phenol content, do the authors used some modification to account for the contribution of vitamin C?
Regarding the total flavonoid content, emmploying a protocol with NaNO2 provides information for non-flavonoid compounds as well. Please adivise: Papoti, V. T., Xystouris, S., Papagianni, G., & Tsimidou, M. Z. (2011). " Total flavonoid" content assessment via aluminum [AL (III)] complexation reactions. What we really measure?. Italian Journal of Food Science, 23(3), 252. and recent literature on the matter
furthermore, the authors mention that as standard QE was used (I suppose quercetin), then the results are expressed as GAE, euqual to total phenols and then, in the footnote of Table 3 the values are reported as catechol equivalents (?)
As the validity of results is a mmatter of correct methodology, I suggest a better description (detailed) of the analytical part.
2.8 HPLC, the authors have determined a wide array of commpounds that absorb at different maxima. Nevertheless a single wavelength is reported, the general one to monitor phenols and there is no information on the quantification process, regarding the reference/s compound/s used, the standard/s curve/s etc.
In Table 3 at least, after th optimization does not indicate a significant change (or improvement), whereas differences were observed in Table 4.
In the same Table and in the text, benzoic acid is not a phenolic compound but and organic acid! It should be removed from the table and discussed elsewhere (although organic acid analysis is missing). Compounds like gallic, resveratrol, rutin and myricetin (or myricitrin?)
In figure 4 no clear difference is observed. Error bars are missing.
l.518-521, the differences in pH values and titratable acidity are small to mmy opinion in order to have an oganoleptic impact, and probably what is stated in l. 521-523 is ore possible for the explanation.
In figure 7, something is missing from the legend
Conclusions:l. 528 and 529 are out of the scope according to the aim and previous comments made above
In the introduction, make better screening of the literature to support further the innovation of the work
e.g. Liu, Y., Sheng, J., Li, J., Zhang, P., Tang, F., & Shan, C. (2022). Influence of lactic acid bacteria on physicochemical indexes, sensory and flavor characteristics of fermented sea buckthorn juice. Food Bioscience, 46, 101519.
Author Response
Response to Reviewer 3 Comments
Comments and Suggestions for Authors
present paper describes the Impact of fermentation of sea buckthorn juice with L. plantarum on antioxidant, antimicrobial, antihypertension and anticancer activity
- considering the aforementioned title and the lines 68-72 the reader does not expect to find in the results and discussion section the parts 3.1 and 3.2 which refer to the berries. Thus, it is redundund and out of the scope of the work. To support that it is the characterization of the raw material, other commmplementary analyses as those mmade in the juice would be required.
Response: As suggested the modification have been made in revised manuscript and the title of the manuscript has been modified to be mor informative ““Nutritional evaluation of Sea Buckthorn “Hippophae goniocarpa” berries and the pharmaceutical potential of the fermented juice”.
- A further observation regarding the particular part is that the authors separate the amino acid content from chemical composition(?), whereas in the chemical composition section they put together nutrients and minerals. All the section should have been combined in one paragraph as Nutritional analysis (and within the text to coment on non-essential aino acids)
Response: The suggestion of very acceptable and all nutritional analyses were combined
- The authors comment also on % of daily requirements. Maybe I misunderstood but is there somewhere a daly intake of berries or what is considered as a portion for consummption?
Response: A daily requirement (Daily intake) is referred to the daily intake of every nutrient in the See buckthorn berries not for the whole barriers
- In 2.7 methodology
Regarding the total phenol content, do the authors used some modification to account for the contribution of vitamin C?
Response: Vitamin C was determined as a separate functional nutrient, but it was not included in the total phenolic compounds because it is not a phenolic compound. On the other hand, the determination of antioxidant activity is including all nutrients which has an antioxidant power including vitamin c.
- Regarding the total flavonoid content, emmploying a protocol with NaNO2 provides information for non-flavonoid compounds as well. Please adivise: Papoti, V. T., Xystouris, S., Papagianni, G., & Tsimidou, M. Z. (2011). " Total flavonoid" content assessment via aluminum [AL (III)] complexation reactions. What we really measure?. Italian Journal of Food Science, 23(3), 252. and recent literature on the matter
Response: The method of NaNO2, AlCl3・6H2O and NaOH is a common and widely used method for determining of total flavonoids. The provide article is regarding to the determination of total flavonoids in Olive leaves extract.
- Mengfei Li, Paul W. Pare, Jinlin Zhang, Tianlan Kang, Zhen Zhang, Delong Yang, Kepeng Wang and Hua Xing, 2018. Antioxidant Capacity Connection with Phenolic and Flavonoid Content in Chinese Medicinal Herbs. Rec. Nat. Prod. 12:3 (2018) 239-250.
- Luís M. Magalhães & M. Inês G. S. Almeida & Luísa Barreiros & Salette Reis & Marcela A. Segundo. Automatic Aluminum Chloride Method for Routine Estimation of Total Flavonoids in Red Wines and Teas. Food Anal. Methods (2012) 5:530–539.
- Özge Yıldız1, Beyza VahapoÄŸlu, Mehmet Ali Marangoz, Esra ÇapanoÄŸlu Güven, Alev Bayındırlı. Determination of Phenolic Compound Profiles and Antioxidant Effect of Plant Extracts on Late-Release Soft Lozenge. EAS J Nutr Food Sci., Volume-3 | Issue-6 | Nov-Dec; 2021.
- furthermore, the authors mention that as standard QE was used (I suppose quercetin), then the results are expressed as GAE, euqual to total phenols and then, in the footnote of Table 3 the values are reported as catechol equivalents (?)
Response: The total phenolic content is calculating as GAE but the total flavonoids calculating as QE. As suggested the modification have been made in revised manuscript.
- As the validity of results is a matter of correct methodology, I suggest a better description (detailed) of the analytical part.
Response: The descriptions of the analytical methods have been stated in the methodology to be repeatable.
- 8 HPLC, the authors have determined a wide array of compounds that absorb at different maxima. Nevertheless, a single wavelength is reported, the general one to monitor phenols and there is no information on the quantification process, regarding the reference/s compound/s used, the standard/s curve/s etc.
Response: The HPLC analysis was conducted against the known standard of the phenolic and flavonoids mix. The type and concentration of every compound was determined according to the retention time and the area under the peak of every compound verses the reference.
- In Table 3 at least, after the optimization does not indicate a significant change (or improvement), whereas differences were observed in Table 4.
Response: In the table 3, the significant difference could be noted in total phenolic content, total flavonoids, vitamin C and sulphated polysaccharides (Mean values in a line having different superscript are significantly different at (p ≤ 0.05). (p ≤ 0.05), the same in table 4, we can see the significant increase in the phenolic compounds. So, the significant improvement has been determined in the fermented juice.
- In the same Table and in the text, benzoic acid is not a phenolic compound but and organic acid! It should be removed from the table and discussed elsewhere (although organic acid analysis is missing). Compounds like gallic, resveratrol, rutin and myricetin (or myricitrin?)
Response: Phenolic acids are major components of berries. Phenolic acids are represented by cinnamic and benzoic acid derivatives. Benzoic acid derivatives include p-hydroxybenzoic acid, salicylic acid, gallic acid, and ellagic acid. So, the benzoic acid is an organic acid and precursor of phenolic acids.
- In figure 4 no clear difference is observed. Error bars are missing.
Response: The HPLC analysis was conducted one time analysis, it was very difficult to conduct the HPLC analysis 3 times to present a standard deviation or standard error (it is costly to conduct the HPLC analysis 3 times)
- 518-521, the differences in pH values and titratable acidity are small to my opinion in order to have an oganoleptic impact, and probably what is stated in l. 521-523 is ore possible for the explanation.
Response: It is true, the titratable acidity its effect on organoleptic properties has been reformatted and modified in a discussion.
- In figure 7, something is missing from the legend
Response: A full explanation of the figure 7 was added
- Conclusions: l. 528 and 529 are out of the scope according to the aim and previous comments made above
Response: As suggested the modification have been made in revised manuscript.
- In the introduction, make better screening of the literature to support further the innovation of the work
Response: The introduction has been improved accordingly.

Round 2
Reviewer 1 Report
good work and it should be accepted. best of luck...
Author Response
Response to reviewer 1 comments Comments and Suggestions for Authors Comment: good work and it should be accepted. best of luck...Response: Thank you for your effort and time to improve the quality of the manuscript
Reviewer 3 Report
Dear Authors,
despite the fact that the manuscript has been improved there are still some issues that were not answered. Maybe i did not ake myshelf clear.
Vitamin C as you mentioned has been determined separately, however it interfere with the determination of total phenols and cause overestimation. Although this is known for years, see a recent article
Roslan, A. S., Ando, Y., Azlan, A., & Ismail, A. (2019). Effect of Glucose and Ascorbic Acid on Total Phenolic Content Estimation of Green Tea and Commercial Fruit Juices by Using Folin Ciocalteu and Fast Blue BB Assays. Pertanika Journal of Tropical Agricultural Science, 42(2).
Thus, F-C in such products rather determine the reducing (antioxidant activity) capacity of the sample than the total phenol content. So I would rather present it as an additional antioxidant assay.
regarding the total flavonoids, to mention a set of publications using the particular assay does not necessarily justify its use. Did they search the methodology or relevant literature? The paper I provided examined the methodology before applying any protocol for analysis and comment on the role of NaNO2. to make myself clear check the paper (AliciaGil-Ramírez et al., 2016 https://doi.org/10.1016/j.jff.2016.05.005) who comment on the total flavonoid content findings on mushroomms in various publications using chromogenic assays explaining that genetically mushrooms do not biosynthesize flavonoids! Thus, after understanding on what exactly is measured, then i would write more appropriately what was determined.
Regarding HPLC, there is no mention on the mixed standard used for quantification. In addition quantification usually is (if not at exactly the maximum of each commpound) carried out at different wavelengths depending on whether the target compounds are hydroxybenzoic acids, hydroxycinnammic acids or flavonoids. In addition, you keep in the Table as phenol the value of benzoic acid, which is not a phenol (and it is not justifyiable that it is a precursor of hydroxybenoic acids). After all, i recommended to use the values somehow in the text.
The fact that HPLC is costly is not a reason for a single analysis. You can check repeatability and do dublicate. Otherwise comment more on qualitative differences. In paragraph 3.4 you provide statistics for the HPLC results (ANOVA). How was this feasible when you have a single value.
In figure 4, legent, it is DPPH and not DDPH
Author Response
Response to the reviewer 3 Comments
Comments and Suggestions for Authors
Dear Authors,
despite the fact that the manuscript has been improved there are still some issues that were not answered. Maybe i did not ake myshelf clear.
- Vitamin C as you mentioned has been determined separately, however it interfere with the determination of total phenols and cause overestimation. Although this is known for years, see a recent article
Roslan, A. S., Ando, Y., Azlan, A., & Ismail, A. (2019). Effect of Glucose and Ascorbic Acid on Total Phenolic Content Estimation of Green Tea and Commercial Fruit Juices by Using Folin Ciocalteu and Fast Blue BB Assays. Pertanika Journal of Tropical Agricultural Science, 42(2).
Reply: Thanks you for your valuable comment, I would like clarify, in methodology of Vitamin C was estimated in sea buckthorn juice using a titrimetric method by 2, 6-di-chloro-phenol-indophenol reagent and this official methods of Vitamin C according to AOAC Official Methods of Analysis 45.1.14 (AOAC Method 967.21) has been recommended for the analysis of Vitamin C in beverages and juices for the purpose of nutrition labelling. Also, this method used to determination of Vitamin C and published in highly reputed journals. Also, the vitamin C value is in between the normal range of the sea buckthorn determination in other research papers
Thus, F-C in such products rather determine the reducing (antioxidant activity) capacity of the sample than the total phenol content. So I would rather present it as an additional antioxidant assay.
Reply: Thanks you for your valuable comment, I would like to explain that the determination of the antioxidant activity by two methods (DPPH and ABTS) for SBJ: Sea buckthorn juice; SBJ-Pl: Sea buckthorn juice fermented with Lactobacillus plantarum present in figure 4 in the manuscript rather than phytochemical and physical analysis of sea buckthorn juice containing total phenolics and flavonoids that present in table 3.
So, I have added the statement in the discussion “Here, it should be mentioned that the existence of vitamin C may interfere in the measuring of the total phenic and flavonoids content resulted to raising in of their content in the juice but in all cases it is showing the increasing of antioxidant capacity of the fermented juice”
- regarding the total flavonoids, to mention a set of publications using the particular assay does not necessarily justify its use. Did they search the methodology or relevant literature? The paper I provided examined the methodology before applying any protocol for analysis and comment on the role of NaNO2. to make myself clear check the paper (AliciaGil-Ramírez et al., 2016 https://doi.org/10.1016/j.jff.2016.05.005) who comment on the total flavonoid content findings on mushroomms in various publications using chromogenic assays explaining that genetically mushrooms do not biosynthesize flavonoids! Thus, after understanding on what exactly is measured, then i would write more appropriately what was determined.
Reply: Thanks you for your valuable comment, Sodium nitrate is added with AlCl3, which contributes to a color complex compounds, measured at wavelength 510, in alkaline pH by add NaOH. Also, this method used to determination of total flavonoids and published in highly reputed journals.
For example: - Rosario Goyeneche, Sara Roura, Alejandra Ponce, Antonio Vega-Gálvez, Issis Quispe-Fuentes, Elsa Uribe, Karina Di Scala, Chemical characterization and antioxidant capacity of red radish (Raphanus sativus L.) leaves and roots, Journal of Functional Foods,16, 2015, 256-264, https://doi.org/10.1016/j.jff.2015.04.049.
- Regarding HPLC, there is no mention on the mixed standard used for quantification. In addition, quantification usually is (if not at exactly the maximum of each commpound) carried out at different wavelengths depending on whether the target compounds are hydroxybenzoic acids, hydroxycinnammic acids or flavonoids. In addition, you keep in the Table as phenol the value of benzoic acid, which is not a phenol (and it is not justifiable that it is a precursor of hydroxybenoic acids). After all, i recommended to use the values somehow in the text.
The fact that HPLC is costly is not a reason for a single analysis. You can check repeatability and do duplicate. Otherwise comment more on qualitative differences. In paragraph 3.4 you provide statistics for the HPLC results (ANOVA). How was this feasible when you have a single value?
Reply: Thanks you for your valuable comment, HPLC analysis undergo in Faculty of Agriculture central lab which is accreditation certificated lab. This lab depends on previously calibrated techniques so the percent of errors is almost zero. Also, the HPLC analysis is completely automated method so there is no man handling error.
On the other hand, I would like to clarify that the standard curve of phytochemical contains hydroxybenzoic acids, hydroxycinnammic acids, so it appeared among the phytochemical compounds that were determination. So the title of phytochemical table title has been changed from HPLC analysis of phenolic and flavonoids content of sea buckthorn juice to HPLC analysis of organic acids and other phenols of sea buckthorn juice
The original HPLC analysis graphs have been attached
- In figure 4, legent, it is DPPH and not DDPH
Reply: the Typing mistake has been corrected

Round 3
Reviewer 3 Report
Dear authors,
I recommend the acceptance